# Management Food Waste in Municipality Schools: An Analysis from a Circular Economy Perspective

Simone Sehnem [1,2], Lucia Godoi [2], Flavio Simioni [3], Cristina Martins [2], Sandro Vieira Soares [2], José Baltazar Salgueirinho Osório de Andrade Guerra [2] and Tais Provensi [1,*]

1    Postgraduate Program in Management, University of West Santa Catarina (UNOESC), Chapecó 89813-000, Brazil
2    Postgraduate Program in Management, University of South Santa Catarina (UNISUL), Florianópolis 88010-010, Brazil
3    Postgraduate Program in Environmental Sciences, Santa Catarina State University (UDESC), Lages 88520-000, Brazil
*    Correspondence: taisprovensi@gmail.com

**Abstract:** *Background*: Food waste is a situation that triggers certain controversy, considering that there is still a significant number of people who do not have access to healthy and nutritious food every day. The management of food leftovers from school lunches depends on the creation of measurement mechanisms. Thus, current characteristics of schools should be identified in order to mitigate and better manage these leftovers. This study addresses this gap, that is, it seeks to understand how food waste management is carried out in schools from a circular economy perspective. *Methods*: Focusing on the management of school lunches in municipal schools, this study aims to build an inductive interpretive theory in order to understand how schools promote food waste management. *Results*: Loss estimates and their monetization were identified and explained, and alternatives for waste recovery using the ReSOLVE framework were proposed. *Conclusions*: Therefore, it was assumed that the perspective of circularity is a possible and viable way to manage food waste in school lunches. Stakeholder engagement and awareness raising become necessary premises for success in the food circularity journey.

**Keywords:** sustainability; circular economy; food waste management



## 1. Introduction

Producing food demands consumption of natural resources on an exponential scale [1]. Such production yields healthy food, recognized as nutritious and capable of satisfying the hunger of the most diverse species. However, between producing and consuming food, there is a gigantic scale of losses. Even consumption, in which there is desire to have more food than the individual is actually willing to eat, contributes to having leftovers, which are usually thrown away [2]. As such, waste reduction is elementary [3].

In school lunches, there is another aggravating factor, as many schools opt for the "cooked meal", that is, the meal served. Thus, the child does not have the right to choose the types of food they consume, nor the quantities. This fact has a brutal impact on the increase in food waste. However, in order to understand and manage this waste of food, it is essential to focus on measuring it. Measuring, weighing, monetizing school food loss is not a common practice in emerging countries, but it is an important and necessary way to internalize new guidelines for the management of leftovers and waste. This is a way of contributing to the reduction in waste at the source [4].

Food waste and leftovers represent significant sums for the Brazilian economy [1]. Waste refers to the end of the food chain [5], that is, retail and consumption, and leftovers refer to food that is prepared in excess, served and not consumed. According to the special report released by [6] (p. 1), food waste consists of "any product or part of a product

grown, fished or processed for human consumption that could have been consumed if it had been treated or stored differently". Although there is no unanimous definition on the subject, it can be inferred from the various concepts established that food is produced to be consumed; if this objective is not achieved, the factors causing interruption of the purpose are considered food waste.

According to information released by [7] (p. 1), "between a quarter and a third of the food produced annually for human consumption is wasted, which would be enough to feed two billion people". Therefore, it is not possible to admit that food in conditions of consumption ends up going to waste. Waste is related to consumption, which is the focus of this work. Waste can be observed in the following situations: leftovers left on the plate; product processed but not distributed; spoiled food; and waste during storage [8].

The following items can be added to the list of factors that cause waste: poor planning of the forecast of demand for meals; food preferences; qualification and training of employees; absence of quality indicators; purchases made without criteria; and climate [9]. There are three predominant waste factors: the correction factor; surplus; and leftovers. The correction factor corresponds to weight loss in relation to the initial weight due to the removal of inedible parts at the time of food preparation. Surpluses are food produced but not distributed. Leftovers refer to food distributed but not consumed [10]. Waste control is a key factor in detecting practices that increase waste; as such, it is necessary to correctly plan the number of meals per day apply goals so that the entire team is committed to controlling leftovers, training the team and making the dish presentable [11].

Raising awareness on the topic has national and international repercussions, resulting in several initiatives. One of these is Save Food, a program that was born from the National Committee for the Reduction of Food Loss and Waste and has the support of the national representation of the United Nations Food and Agriculture Organization [12], organized by the government, civil society, researchers and the private sector, whose mission is to encourage the creation of policies and strategies and organize global communication campaigns [13]. In order to achieve the objective of generating actions to reduce food waste, it is necessary to identify the quantities of waste produced [14]. It is important to quantify wasted food and correlate it with the factors that induce consumers to waste food [15].

The three main pillars of food service are as follows: (a) the environment, which involves all aspects related to the perception of the place; (b) service, which must show efficiency, organization and be appropriate to the restaurant's concept; and (c) menu, which must meet the expectations of the target audience [10].

Without monitoring mechanisms, there is no reliability or credibility in the quality of the service. As such, it is necessary to provide the following information: leftover intake; dirty leftovers; and clean leftovers [12]. These concepts are related to waste in Food and Nutrition Units [16]. The definition of dirty leftovers is "the food that is left in the vats of the distribution counter after the end of the meals and that should not be reused", whereas the definition of clean leftovers is "food that was prepared, but not distributed, and must be kept under refrigeration with time/temperature monitoring" [16] (p. 2040).

To introduce efficient alternatives for managing food waste, the circular economy offers a set of business models, such as 10Rs, the ReSOLVE framework and circular inputs for optimization and efficiency in food use. Studies that point out the role of the circular economy in generating alternatives for resource recovery and reintroduction into new production cycles has been performed, with emphasis alludes to the valorization of food processing with adherence to the assumptions of the circular economy [17], description food waste recovery alternatives and their economic and environmental implications [18], and recovery of food waste from a circular economy perspective [5].

Considering this context, this study aims to answer the following research question: How can food waste be managed in schools from the circular economy perspective? This question raises the following research objective: To analyze how food waste management is carried out in schools from the circular economy perspective.

This study is justified and supported by some indicators that inform that families waste, on average, 353 g of food per day or 128.8 kg per year [19], which signals environmental irresponsibility [20]. Waste, in this sense, is equivalent to 114 g daily per capita, representing annual waste of 41.6 kg per person. These values only take into account waste resulting from homemade food [19]. In schools, these indicators have not yet been mapped nor measured. This shows an opportunity for a practical and applied contribution to the school environment, to what can be called proactive corporate social responsibility [21], specially creating indicators to manage the food waste reduction [22]. The dissemination of information benefits the reduction and waste of food [23].

In addition to this introduction, the structure of this paper includes a section that deals with methodological procedures. In the sequence, the research data are presented and analyzed, followed by a discussion of the results. Finally, the final considerations of the study are presented, followed by the references.

## 2. Materials and Methods

This study was carried out in municipal schools in Florianópolis, state of Santa Catarina, Brazil. The municipal schools are located in five regions of the city: thirteen in the north of the island; seven in the eastern region; nine in the central region; one in the continental region; and eight in the south of the island, totaling 38 municipal schools in the city. The number of enrollments available only for elementary schools was 14,531, a number estimated in the municipal education plan, which was released in November 2014 and is no longer updated on the municipal government website [24].

The justification for choosing these units of analysis was associated with ease of access and because they represent the desired profile for this research, namely public municipal elementary education institutions. They offer early childhood, high school and youth and adult education. Of the 38 schools mapped, 17 school unit directors agreed to participate in the research, as shown in Table 1.

**Table 1.** Individuals approached in the research.

| Individuals Approached | No. |
|---|---|
| Principals of school units invited to participate in the research | 38 |
| School unit directors who agreed to participate in the research (semi-structured interviews) | 17 |
| Lunch workers from each school invited to participate in the survey (questionnaire) | 38 |
| Lunch workers who agreed to participate in the research (questionnaire) | 7 |
| Nutritionist responsible for 12 educational units agreed to participate (semi-structured interview) | 1 |
| Direct advisor to the person responsible for school meals in the municipality that agreed to participate (interview) | 1 |

Subjects occupying different positions were chosen to provide a set of data that could fully support the research. It is understood that the experience of these professionals allows them to provide relevant and strategic knowledge for understanding the dynamics of managing school lunches, their leftovers and waste. These agents who know the school reality can be considered the key informants of the research as they hold strategic knowledge about the units of analysis. In this way, it was possible to adopt the inductive interpretive theory, which follows the premises of [25]. Through this route, it was possible to obtain theoretical insights that narrow the interface between theory and organizational practices of school units. A similar path has already been adopted in other studies, such as [26] and [27]. To complement data collection, the respondents also took pictures, illustrating through images the amount of food waste accumulated during preparation, during the

meal and after the meal. This information supported the monetization of food waste. This methodological approach became suitable for the research problem proposed in this study, given that previous scientific publications support alternatives for the recovery of food waste. In addition, the locus of analysis, i.e., school units, is still little explored to subsidize avenues of management of leftovers and food waste. Considering the relevance of food waste management, schools are teaching and learning spaces and can be a particularly revealing scenario of organizational behaviors and routines.

The premises of the inductive approach were adopted, starting from a broad scope of research to understand the social phenomenon investigated, that is, to promote viable alternatives for the management of food waste. Data were collected and analyzed, following the interpretative path of [27]. Triangulation of data from different sources was also adopted to increase the reliability of the research findings.

The steps followed to conduct the research consisted of:

(a)  An extensive literature review to understand previous publications in the investigated field that allowed the elaboration of a scientific publication [28].
(b)  Documentary research to learn about technical, practical and instrumental aspects associated with the phenomenon studied [29].
(c)  Semi-structured interviews conducted for immersion into the investigated phenomenon [29]. At this stage, we sought to obtain theoretical saturation, that is, the phenomenon investigated was satisfactorily explained by the data obtained [30]. In this study, we sought to obtain thematic saturation, according to the premises advocated by [31], which occurs when a set of sufficiently rich and stable concepts are discovered. To strengthen the research findings, a search was also carried out using diversified sources, namely observation, application of a questionnaire and secondary data, suggesting empirically diverse, significant and theoretically stable evidence.

Data collection took place from October 2021 to January 2022. The script of semi-structured interview questions comprised aspects associated with the following:

(a)  School profile;
(b)  Types of food served in school lunches;
(c)  Ways of accessing food;
(d)  Types of leftovers and food waste;
(e)  Existence of measurements of waste and leftovers;
(f)  Moments in which waste and leftovers occur.

The questionnaire script included information alluding to the following topics:

(a)  Type of meals served;
(b)  Frequency of meals;
(c)  Characteristics of meals;
(d)  Amount of wasted food;
(e)  Systematic management of food waste and leftovers.

The observation script sought to identify images that portrayed the profile of leftovers and food waste in the preparation, meal and post-meal stages (overcooked). The secondary data were focused on mapping financial indicators alluding to school lunches and were collected directly from the Department of Education website and official documents made available by the schools.

Table 2 presents the profile of the subjects interviewed.

**Table 2.** Individuals approached in the research.

| Codename | Scholarity | Position | Years of Service | Interview Duration | Interview Transcript Pages |
|---|---|---|---|---|---|
| E1 | Postgraduate | Director | 4 years | 32.03 | 7 pages |
| E2 | Postgraduate | Director | 1 year | 13.37 | 4 pages |
| E3 | Postgraduate | Director | - | 24.21 | 5 pages |
| E4 | Postgraduate | Director | 4 years 8 months | 16.20 | 3 pages |
| E5 | Graduate | Director | 20 years | 16.09 | 3 pages |
| E6 | Postgraduate | Director | 15 years | 18.31 | 3 pages |
| E7 | Postgraduate | Director | - | 10.40 | 2 pages |
| E8 | Postgraduate | Director | 4 years | 38.26 | 6 pages |
| E9 | Postgraduate | Director | 10 years | 22.12 | 4 pages |
| E10 | Postgraduate | Director | 11 years | 25.14 | 6 pages |
| E11 | Postgraduate | Director | - | 16.22 | 4 pages |
| E12 | Postgraduate | Director | 16 years | 18.15 | 4 pages |
| E13 | Postgraduate | Director | 15 days | 10.05 | 2 pages |
| E14 | Graduate | Director | 5 years | 16.59 | 3 pages |
| E15 | Postgraduate | Director | 8 years | 9.45 | 3 pages |
| E16 | Postgraduate | Director | 5 years | 21.19 | 4 pages |
| E17 | Graduate | Director | 2 years | 10.14 | 3 pages |
| E18 | Graduate | Nutritionist | 3 years | 16.43 | 3 pages |
| E19 | Graduate | DEPAE | 3 years | 23.05 | 3 pages |
| | | | Total | 6 h 28 | pages |

The interviews were transcribed in full. Although a basic interview script was followed, the style of the interviews was flexible and open so that it was possible to maintain and include topics relevant to specific subjects [25]. However, a range of key and basic questions was maintained so that they could contribute to a basic level of scope and consistency of scientific investigation. In addition to triangulating data from different sources, several steps were taken to ensure the quality and reliability of our research. Well-known criteria of credibility, transferability, reliability and confirmability were adopted, namely [32]:

(a)  Reliability criterion: focused on plausible findings and information extracted from primary data, that is, original information from the research participants. Measures were taken to ensure the correct interpretation of the participants' original opinions, such as close involvement with the research field, triangulation of data and methods, emphasis on capturing data and worldviews of various professionals working in the field, locus of analysis, consultation of secondary data and documents, and primary data and interviews. The assumptions of [27] were adopted, along with the sequential analysis of primary and secondary data so that a comparison of insights in all data sources could be performed. Investigator triangulation was also used during data coding, analysis and interpretation of the data, which was followed by inter-pair debriefing. An Excel spreadsheet was used to ensure reliable data storage and management.

(b)  Transferability criterion: to identify the extent to which data can be transferred to other contexts or settings with other respondents and still be considered informative and pragmatically useful. Care was taken to write about the "what", that is, the description of the organization of interest and its choices, but also the "how" of the phenomenon,

that is, emphasis on multiple ways of doing things related to filling in gaps, such as information about the repertoire of research insights that the researcher wants to fill when consulting professionals, as well as different scenarios, contexts and specificities of the object of analysis. The purpose of this was to obtain sufficient information about the research context to make the data and elements discovered meaningful to the research context and other organizational contexts, such as the investigated phenomenon. This allows for the application of the readers' own judgments about transferability to contexts more familiar to the one described in this paper [33].

(c)   Reliability and confirmability criteria: in this item, we sought to investigate how stable the findings are, as well as to ensure to what extent the conclusions are based on the data and not produced by the visions and imagination of the researchers. To meet this purpose, all schools were invited. Only 17 agreed to participate in the study. With the data in hand, intentional sampling was carried out (to control variation and ensure the emergence of stable and saturated central themes). All interviews were conducted face-to-face synchronously, using online tools such as Meet and Zoom. The transcripts were kept in the informant's own words and style. All respondents were given aliases. The interviews were categorized into higher-order analysis categories and themes to yield ongoing verification of interpretations. Topics that could be supported with data from at least two informants and/or secondary data sources were included, as recommended by [34]. Furthermore, reflective discussions were carried out among the researchers to reach an agreement, that is, an alignment regarding the meaning of data and themes. A final verification of the collected data and the management of the data to conduct the analyses were necessary to attest to the reliability of the research.

Finally, food waste projections were made. As there is no weighing measurement in schools, only approximate estimates, we adopted the assumptions of the studies by [1] and [35] to project estimates of food waste in the schools surveyed. Other studies performed the estimation of losses in previous studies, such as that conducted by the authors of [36], who measured waste along the production and distribution chain [12]; who evaluated waste in a university restaurant [37]; who addressed waste in a restaurant chain; [38], who carried out the measurement in a teaching network; and [14], who addressed waste in a municipality of Switzerland.

## 3. Results

Table 3 presents the profile of food waste in the schools surveyed. All schools surveyed serve set meals, that is, the cook prepares the portion and types of food that are be served to each child. Therefore, the child does not have the autonomy to serve themselves, nor choose the type of food and the amount they are willing to consume for lunch. In addition, all schools serve school lunches in morning and afternoon shifts, thus two meals are served per day. During the pandemic, school management sent snacks to students at their homes. This service was offered in delivery format. Therefore, there was access to school lunches during the pandemic in the form of a home office. As soon as it was possible to return to classes, the traditional format of offering food to students in the schools surveyed was maintained. Such data were included in the paper for clarification purposes.

**Table 3.** Profile of food waste in the schools surveyed.

| Codiname | Average Meals/Day | Day/Week on Which Meals are Served | Relevant Attributes + | Focus * | Separation of Leftovers | Destination of Leftovers ** | Waste in Preparation ++ |
|---|---|---|---|---|---|---|---|
| E1 | 644 | 5 | H | 1 | Yes | 5 | NW |
| E2 | 562 | 5 | F | 1 | No | 5 | NW |
| E3 | 523 | 5 | H | 1 | No | 5 | NW |
| E4 | 506 | 5 | F | 1 | No | 5 | G, V |
| E5 | 690 | 5 | H | 1 | Yes | 5 | V, F |
| E6 | 108 | 5 | H | 1 | Yes | 5 | NW |
| E7 | 649 | 5 | H | 1 | No | 5 | G, F |

+ T: tasty; H: healthy; and F: Fresh. * I have enough food in the educational unit to prepare = 1. I do not have much food in the educational unit to prepare = 2. Food is easy to prepare = 3. The food meets students' preferences = 4. ** Intended for the Mesa Brasil Program = 1. Reuse next day = 2. Intended for animal feed from rural producers = 3. Donation to employees and others = 4. Destination for landfill = 5. Reusable material destined for recycling = 6. Other options = 7. ++ G = greenery; V = vegetables; F = fruits; NW = I have no waste.

Table 3 shows that fresh and tasty food were some of the characteristics mentioned by the respondents as being attributes of the food served during school meals. All schools mentioned that they send their leftovers to landfills. This is important evidence that signals that, for now, there is no policy regarding a better destination for food leftovers and waste; the landfill is the only destination. It is important to remember that in Brazil, legislation regarding food safety is strict and good manufacturing and food management practices are demanded in all food sectors. However, there are also more efficient alternatives for forwarding leftovers, such as the Mesa Brasil Program and donations to people in situations of social vulnerability.

Table 4 presents the profile of food waste during food preparation, during the meal and post-meal.

**Table 4.** Profile of food waste during preparation, during the meal and post-meal.

| Items | In the Preparation of Meals * | | | | | | | In the Meal * | | | | | | | After-Meal (Leftovers in the Pan) * | | | | | | |
|---|---|---|---|---|---|---|---|---|---|---|---|---|---|---|---|---|---|---|---|---|---|
| | 1 | 2 | 3 | 4 | 5 | 6 | 7 | 1 | 2 | 3 | 4 | 5 | 6 | 7 | 1 | 2 | 3 | 4 | 5 | 6 | 7 |
| Green | E6 | E5 E1 | E4 | E7 | | | | E6 | E1 E3 E4 E5 E7 | | | | | | E7 | E4 E5 | | E3 | | | |
| Vegetables | E6 E3 | E5 E1 | E7 E5 | | | | | E6 | E1 E2 E3 E4 E7 | | | | | | | | | E3 E4 | | | |
| Fruits | E6 | E5 E1 | E7 | | | | | E4 E5 E6 E7 | E1 E2 E3 | | | | | | E4 E5 | | | E3 | | | |

**Table 4.** *Cont.*

| Items | In the Preparation of Meals * | | | | | | | In the Meal * | | | | | | | After-Meal (Leftovers in the Pan) * | | | | | | |
|---|---|---|---|---|---|---|---|---|---|---|---|---|---|---|---|---|---|---|---|---|---|
| | 1 | 2 | 3 | 4 | 5 | 6 | 7 | 1 | 2 | 3 | 4 | 5 | 6 | 7 | 1 | 2 | 3 | 4 | 5 | 6 | 7 |
| Meat | E6 E5 E4 | E7 E1 | | | | | | E4 E5 E6 E7 | E1 E3 | | | | | | E3 E4 E5 | | | | | | |
| Carbohydrates (rice, potato, cassava, etc.) | E6 E5 | E4 E1 | E7 | | | | | E6 | E1 E3 E4 E5 | E7 | | | | | E4 E7 | E5 | | | | | E6 |
| Bones | E7 E6 | | E5 | E2 | E1 | | E3 | E1 E3 E6 E7 | | | | | | | | | | | E2 | E3 | E1 |
| Shells | E6 E4 | | E5 | E7 E2 | E1 | | E3 | E1 E3 E4 E6 | E7 | E5 | | | E2 | | E4 | | | | E2 | E3 | E1 |
| Seeds | E6 E4 | E5 | E7 | | E1 | | E3 | E1 E3 E4 E6 E7 | E5 | E2 | | | | | E4 | | | | | E3 | E1 |
| Beverages (juice, soda,) | E7 E6 E5 E4 E1 | | | | | | | E1 E3 E4 E6 E7 | | | | | | | E3 | E4 | | | | | |
| Others | | | | | | | | | | | | | | | | | | | | | |

* Leftovers on the plate, on average, per day by the person who has lunch at the educational unit; the serving spoon portion is equal to 50 g. 1 = There is no waste 2 = Less than one dish per day. 3 = 1 to 5 medium dishes per day. 4 = 6 to 10 medium dishes per day. 5 = 11 to 15 medium dishes per day. 6 = 16 to 20 average dishes per day. 7 = More than 21 average dishes per day.

Table 4 clarifies that most cooks do not understand that all unused leftovers, such as stalks, leaves, bark and seeds that are not used for meals, are a type of food waste. It is possible to make this statement based on the information presented in Table 4, in addition to the fact that of the seven questionnaires completed, the cooks claimed in in four of them that there is no waste during preparation. Participants E1 and E3 did not fill out the part of the questionnaire that mentioned waste in food preparation, evidencing their lack of clarity about the process they perform when preparing school meals. Another possible explanation for this is that the respondent felt insecure answering the questionnaire, even though data confidentiality was ensured.

The perception of this lack of awareness on the part of the cooks corroborates the findings in [11,39], when stating that food waste can occur through the practice of discarding food suitable for human consumption or through consumer negligence, thus influencing food insecurity and consuming scarce financial and natural resources. This highlights an important gap that can be filled with training for cooks, creative nutritionists, differentiated and nutritious recipes and awakening in the children of the world regarding the possibilities that food provides for a nutritious and healthy diet.

When it comes to waste during meals, Table 4 shows that similar evidence about food waste was mapped at the time of preparation. In this item, however, there were cooks who did not fill in all the fields of the questionnaire, such as, for example, E2. Another piece of

evidence was the report by interviewee E6 when claiming that there is no waste during meals, something that is practically impossible to happen. On the other hand, the activities of E6′s school were completely remote by the time this research was carried out, which may have caused such a response. This hypothesis can be identified by the pandemic experienced in 2021 during the data collection period. According to [40], the generation of waste depends on cultural factors, and cities play a crucial role in maintaining food, ratifying, in this case, the issue of education and awareness of the most diverse social classes.

After the meal, there was a greater emphasis on the waste of vegetables, fruits, meats and carbohydrates. Table 5 summarizes the profile data and the total amount of food waste, mapped in the schools surveyed.

**Table 5.** Food waste profile in school meals.

| Codename | Monthly Amount * | Destination of Leftovers | Common Types of Waste | Amount of Food Waste in Preparation | Amount of Food Waste in the Meal | Monetary Representation of School Food Waste ** |
|---|---|---|---|---|---|---|
| E1 | BRL 4865.62 | Sanitary landfill; the school neighbor takes the waste to feed animals | Salads | It was never measured by the management report | Two full buckets | Considering an average of 75 dishes, there is daily waste in meals, valued at BRL 180.00. This corresponds to approximately 20 plates full of food wasted per day, stored in two full buckets for final disposal. |
| E2 | BRL 10,843.47 | Landfill | Salads | It was never measured by the management report | On average, 50 g per student | Considering an average of 544 dishes, daily waste is equivalent to BRL 489.60. |
| E3 | BRL 7412.22 | Composter, school garden | Fish (tuna) | It was never measured by the management report | On average, 50 g per student | Considering an average of 315 dishes, daily waste equals BRL 283.50. |
| E4 | BRL 8406.52 | Composter | Salads and Porridge | It was never measured by the management report | On average, 50 g per student | Considering an average of 506 dishes, daily waste equals BRL 455.40. |
| E5 | BRL 9788.88 | Landfill | Porridge and tuna | It was measured at some point, but the data were lost in the management report | On average, 50 g per student | Considering an average of 644 dishes, daily waste is equivalent to BRL 579.60. |

**Table 5.** *Cont.*

| Codename | Monthly Amount * | Destination of Leftovers | Common Types of Waste | Amount of Food Waste in Preparation | Amount of Food Waste in the Meal | Monetary Representation of School Food Waste ** |
|---|---|---|---|---|---|---|
| E6 | BRL 11,650.30 | Composter | Salad, tuna | It was never measured by the management report | On average, 50 g per student | Considering an average of 562 dishes, daily waste is equivalent to BRL 505.80. |
| E7 | BRL 4672.80 | Landfill | Fruits and vegetables | It was never measured by the management report | On average, 50 g per student | Considering an average of 649 dishes, daily waste is equivalent to BRL 584.10. |
| E8 | BRL 16,351.98 | Landfill | Food in general | It was never measured by the management report | On average, 50 g per student | Considering an average of 819 dishes, daily waste is equivalent to BRL 737.10. |
| E9 | BRL 7996.00 | Landfill | Food in general | It was never measured by the management report | On average, 50 g per student | Considering an average of 523 dishes, daily waste is equivalent to BRL 470.70. |
| E10 | BRL 7687.48 | Landfill | Bread | It was never measured by the management report | On average, 50 g per student | Considering an average of 644 dishes, daily waste is equivalent to BRL 579.60. |
| E11 | BRL 11,520.39 | Landfill | Food in general | It was never measured by the management report | On average, 50 g per student | Considering an average of 421 dishes, daily waste is equivalent to BRL 378.90. |
| E12 | BRL 12,834.10 | Landfill | Fruits and vegetables | They were already been measured; there are no data as reports only focused on reducing waste | On average, 50 g per student | Considering an average of 690 dishes, daily waste is equivalent to BRL 621.00. |
| E13 | BRL 7153.00 | Composter | Vegetables and greenery | They have already been measured; there are no data as reports only focused on reducing waste | On average, 50 g per student | Considering an average of 444 dishes, daily waste is equivalent to BRL 399.60. |
| E14 | BRL 6423.85 | Landfill | Fruits | It was never measured according to the director's report | On average, 50 g per student | Considering an average of 108 dishes, daily waste is equivalent to BRL 97.20. |

Table 5. *Cont.*

| Codename | Monthly Amount * | Destination of Leftovers | Common Types of Waste | Amount of Food Waste in Preparation | Amount of Food Waste in the Meal | Monetary Representation of School Food Waste ** |
|---|---|---|---|---|---|---|
| E15 | BRL 5607.36 | Landfill | Vegetables and salads | It was never measured according to the director's report | On average, 50 g per student | Considering an average of 649 dishes, daily waste is equivalent to BRL 584.10. |
| E16 | BRL 5607.36 | Landfill | Salads | It was never measured according to the director's report | On average, 50 g per student | Considering an average of 649 dishes, daily waste is equivalent to BRL 584.10. |
| E17 | 9731.98 | Landfill | Food in general | It was never measured, according to the director's report | On average, 50 g per student | Considering an average of 382 dishes, daily waste is equivalent to BRL 343.40. |

* Values informed by the Municipal Education Department. This value corresponds to the total amount passed on monthly by the city government to pay for school meals. ** Values informed by the Municipal Education Department. The simulation considers the estimate of an average of 20 days of class, with an average price of BRL 9.00 per meal. Daily waste of meals is valued at BRL 1.80 per student. This number is justified based on 50 g of waste per student, considering that 50% of students waste.

We present here the details of the steps followed to conduct the monetization of school lunch waste:

- completing a form where respondents indicated the type of food wasted and the amount (in portions).
- estimation of the amount wasted before, during and after the meal.
- knowing the amount in grams, the price of a popular meal was used to estimate costs.
- referring to partial monetization, which refers to partially consuming lunch, that is, eating two items and leaving one on the plate, for example.

The data in Table 5 clearly show that most of the food wasted goes to sanitary landfills, and that these inputs are not reused. This practice confirms the findings of [41], pointing out that food waste is the biggest contributor to landfills, which, in turn, generate methane, a powerful greenhouse gas that directly affects the environment.

Foods that are wasted most often are fruits and vegetables, and there are also reports of non-preference for tuna and porridge. In the perception of the principals, rejection of these foods may be related to eating habits that students have at home. Thus, as they are not used to consuming these types of food, they end up rejecting them in the school environment.

Regarding the measurement of food during preparation, participants E5, E12 and E13 mentioned that they have weighed food waste; however, they do not do this anymore and do not have any formalized data; rather, they emphasize that it was a measure that gave results at the time, that is, waste was mitigated with the practice of weighing. As for food waste, when referring to what was left on students' plates, the reports confirm that this occurs, but there is no accurate measurement. Therefore, an average was established for this calculation, using 50 g per plate, which is equivalent to a serving spoon for each student. In this way, it was possible to estimate food waste in each school.

What drew the most attention in the analysis was the fact that the directors spoke the truth about the real situation, as evidenced in the speech of interviewee E1, who explained that two buckets full of food go to waste every day, demonstrating his indignation with such a situation. On the other hand, interviewee E4 said he does not really pay attention to the food waste process, showing indifference to the situation. On the other, some other

directors were interested in having in school projects that contribute to the preservation of the environment and projects developed with the aim of reducing any type of waste.

As for the monetary estimate, the values are expressive, even though they do not seem like much when measured individually. However, considering the values altogether, the situation becomes more evident and aggravating, requiring urgent changes. The daily amount, based on 50 g of waste on the plate, i.e., equivalent to a tablespoon, and considering that 50% of students waste food, reaches the value of BRL 7876.70 per day, without considering the waste in the kitchen. That is, in one year, this value can reach BRL 157,534.00. If the value is compared with the financial transfer that is made annually to the educational units, it can be estimated that 39.9% of the amount invested is wasted.

The data evidenced in this analysis corroborate the speech of [18], in which it is stated that about a third of food is wasted globally, requiring significant resources for treatment and disposal. It is reinforced that, in this case, there is a waste of valuable resources, because food is an asset that could end hunger but is sent to landfills instead. There was an average estimated waste of 50 g of food during preparation (E1, E2, E3, E5 and E6) and from 101 to 150 g by E4 and E7. The types of food that are most often wasted are greeneries, fruits, vegetables, meat and carbohydrates. In meals, emphasis was given to waste of up to 50 g per meal served in schools reported by E1, E3, E4, E5, E6 and E7, as well as to waste ranging from 51 to 100 g per dish served at the school of E2. Greeneries, fruits, vegetables, meat and carbohydrates are also wasted. After the meal, waste of up to 50 g was highlighted by the schools of E1, E5, E6 and E7. Respondent E3 mentioned that waste is estimated between 51 and 100 g per dish served. E2 reported that waste ranges from 101 to 150 g of per meal served. Juice was reported to be wasted by E4, E5, E6 and E7, while E1, E2 and E3 signaled that they have no such waste. Only cook E5 indicated that she did not waste any fruit or meat. Even though this seems like small amounts, added to the number of students who receive meals, waste becomes quite significant. Table 6 presents the monetization of food waste and leftovers in the surveyed municipal schools.

**Table 6.** Monetization of leftovers and waste meals during preparation, during the meal and post-metal.

| | **Estimation of Waste and Leftovers in Preparation** | | | | | | |
|---|---|---|---|---|---|---|---|
| **Codename** | **Estimated Fruit Leftovers *** | **Estimated Vegetable Surplus *** | **Estimation of Vegetable Leftover *** | **Estimation of Carb Leftover *** | **Estimation of Total Leftovers in kg** | **BRL of Waste per Day** | **Estimated Surplus per Month in BRL *****|
| E1, E2, E3, E4 E5, E6, E7 | Around 1 plate ** BRL 2.25/day 500 g/day | Around 1 plate ** BRL 2.25/day 500 g/day | Around 1 plate ** BRL 2.25/day 500 g/day | Around 1 plate ** BRL 2.25/day 500 g/day | 12,250 kg/day | BRL 63.00 | BRL 1260.00 |
| E8, E9, E10, E11 E12, E13, E14, E15, E16, E17 | Around 1 plate ** BRL 2.25/day 500 g/day | Around 1 plate ** BRL 2.25/day 500 g/day | Around 1 plate ** BRL 2.25/day 500 g/day | Around 1 plate ** BRL 2.25/day 500 g/day | 17,500 kg/day | BRL 90.00 | BRL 1800.00 |
| | Estimation of waste and leftovers in the meal (leftovers on the plate) | | | | | | |
| E1, E2, E3, E4 E5, E6, E7 | Around 1 plate *** BRL 2.25 250 g/day | Around 4 dishes *** BRL 9.00 1 kg/day | Around 7 dishes *** BRL 15.75/day 3.5 kg/day | Around 8 dishes *** BRL 18.00/day 2.0 kg/day | 47,250 kg/day | BRL 315.00 | BRL 6300.00 |

**Table 6.** *Cont.*

| | | | Estimation of Waste and Leftovers in Preparation | | | | |
|---|---|---|---|---|---|---|---|
| Codename | Estimated Fruit Leftovers * | Estimated Vegetable Surplus * | Estimation of Vegetable Leftover * | Estimation of Carb Leftover * | Estimation of Total Leftovers in kg | BRL of Waste per Day | Estimated Surplus per Month in BRL ***** |
| E8, E9, E10, E11 E12, E13, E14, E15, E16, E17 | Around 1 plate *** BRL 2.25/day 250 g/day | Around 2 dishes *** BRL 4.50/day 500 g/day | Around 4 dishes *** BRL 9.00/day 1 kg/day | Around 5 dishes *** BRL 11.25/day 1.25 kg/day | 67,500 kg/day | BRL 270.00 | BRL 5400.00 |
| | | | Estimation of post-meal waste and leftovers (leftovers in the pan) | | | | |
| E1, E2, E3, E4 E5, E6, E7 | Around 2 dishes **** BRL 16.00/day 1 kg/day | around 6 dishes *** BRL 13.50 1.5 kg/day | Around 3 dishes ** BRL 6.75/day 1.5 kg/day | Around 5 dishes ** BRL 11.25/day 2.5 kg/day | 45,500 kg/day | BRL 332.50 | BRL 6650.00 |
| E8, E9, E10, E11 E12, E13, E14, E15, E16, E17 | Around 2 dishes **** BRL 16.00/day 1 kg/day | Around 6 dishes *** BRL 13.50 1.5 kg/day | Around 3 dishes ** BRL 6.75/day 1.5 kg/day | Around 5 dishes ** BRL 11.25/day 2.5 kg/day | 65,000 kg/day | BRL 475.00 | BRL 9500.00 |

* For schools that did not answer this question, the parameter of the first school that answered was used, observing sequence E1 to E17. ** For those calculated, the value of BRL 2.25/plate was considered, and 1 plate = 500 g. *** For those calculated, the value of BRL 2.25/plate was considered, and 1 plate = 250 g. **** For those calculated, the value of BRL 9.00/dish was considered, and 1 plate = 500 g. ***** In the estimate, an average of 20 days of classes was considered, with an average price of BRL 9.00 per dish.

For the preparation and analysis of Table 6, data from the questionnaires were used. Based on the information obtained, it was possible to estimate an average of 250 to 500 g per dish for the seven responding schools. As for the schools that did not fully answer the questionnaire, the parameter of the first school that responded in the sequence of E1 to E17 was used. In this estimate, an average of 20 school days per month was considered, with an average meal price of BRL 9.00 (i.e., price of Bandejão, a popular meal served in the city of Florianópolis). This parameter needed to be estimated to arrive at a monetary value of food preparation. With this estimate defined, a simulation of the economic loss was performed, arising from the leftovers on students' plates (Table 7).

Furthermore, there was a high discard rate during the students' meals. This factor may occur due to a few circumstances, such as the following: (i) the student is not used to some types of food offered; (ii) the way in which the meal is offered; (iii) mixed foods, such as fruit salads; and (iv) in schools, there is no possibility of self service, which would be more in accordance with students' tastes and desired quantities. For this analysis, the same logic observed for Table 6 was followed, that is, an average of dishes was estimated according to the notes of the seven questionnaires completed by the cooks. Thus, it was possible to estimate an average of 250 to 500 g per plate for the seven respondent schools. For schools that did not fully answer the questionnaire, the parameter of the first school that responded was used, maintaining the sequence of E1 to E17. In this estimate, an average of 20 days of classes per month was considered, with an average meal price of BRL 9.00. This parameter needed to be performed to arrive at a monetary value of food preparation. The numbers are conservative and, even so, the final amount of waste is very large.

Emphasis is given to the leftovers after preparing the meal, that is, what is left in the pan, which, in turn, proved to correspond to the highest amounts of waste when compared to waste before and after the meal. It is noteworthy that fruits, when left over, can be stored for another day to be redistributed to the students. The same occurs with vegetables, if they are not seasoned. However, it is not possible to reuse cooked foods, which was highlighted

by the interviewees. They mentioned that they follow the nutritionist's orientation, i.e., to not save prepared food for the next day. Table 7 indicates the amount of waste, in kilograms and in reais, found in previous analyses.

**Table 7.** Total monetization of leftovers from school meals.

| Codename * | | |
|---|---|---|
| E1, E2, E3, E4 E5, E6, E7, | Estimate of surplus in preparation in kg | 12,250 kg/day |
| | Estimate of surplus in preparation in BRL | BRL 1260.00 |
| | Meal leftover estimate in kg | 47,250 kg/day |
| | Meal leftover estimate in BRL | BRL 6300.00 |
| | Estimated surplus in post-preparation in kg | 45,500 kg/day |
| | Estimated surplus in post-preparation in BRL | BRL 6600.00 |
| | Total in Kg and in BRL ** | 105 kg and BRL 14,210.00 |
| E8, E9, E10, E11, E12, E13, E14, E15, E16, E17 | Estimate of surplus in preparation in kg | 17,500 kg/day |
| | Estimate of surplus in preparation in BRL | BRL 1800.00 |
| | Meal leftover estimate in kg | 67,500 kg/day |
| | Meal leftover estimate in BRL | BRL 5400.00 |
| | Estimated surplus in post-preparation in kg | 65 kg/day |
| | Estimated surplus in post-preparation in BRL | BRL 9500.00 |
| | Total in Kg and in BRL ** | 150 kg and BRL 16,700.00 |

* For schools that did not answer this question, the parameter of the first school thar answered was used, observing, however, the sequence E1 to E17. ** In the estimate, an average of 20 days of classes was considered, with an average price of BRL 9.00 per dish.

Table 7 shows the kilograms of food wasted and how much this represents, in monetary terms, for public coffers. In this sense, this analysis was challenging because schools do not have a standard parameter for waste management. Each school operates in the way it believes to be the best to fulfill its demands. The projections made here utilized the information provided by the cooks and nutritionists, which allowed the creation of parameters that could support a possible waste scenario. On-site observation also contributes to generating important insights into the waste that occurs in schools.

The value informed is conservative data because the price of food has increased. This is the perception of consumers when making purchases in the market, as well as those who use vehicle fuel for transport. Thus, it is possible to identify and even prove the increase in costs, which also implicates the increase in the value of food. The Food and Agriculture Organization (FAO) of the United Nations released a study earlier this year which reveals that the average price of food in 2021 was the highest in the last 10 years. The FAO Food Price Index was 28.1% higher than in 2020.

Regarding the interviews carried out, it was possible to map that E5, E7 and E17 indicated the regular existence of leftovers. All other schools highlighted that having waste is more common. All respondents indicated that they do not have a policy to reuse school meals, that is, leftovers in the pan; thus, food ends up going in the trash. Among the foods that have the highest volume of waste, the respondents mentioned mixed salads, porridge, tuna and sardines.

With regard to the reduction in waste, the reports demonstrated that food can also be redistributed; however, this only applies to dry foods. When there is food left in a school, the cook reports it to the nutritionist, who participates in a WhatsApp group that covers all school units. In this group, it is possible to communicate surpluses and what food is lacking. However, when it comes to perishable foods, such as fruits, that are identified to spoil, the cooks put the fruits on a display and make them available to students to pick up

and take home. In this sense, excerpts from the report of interviewees E4 and E6 highlight such a statement.

> [ . . . ] what's left of what we get, she (nutritionist) has a group, she's always asking, do you have chicken, to carry out the redistribution of food between school units. So, she always, almost every day asks . . . So that there is not much left to reuse, if there is leftover on the menu, she doesn't send it anymore.
>
> (Interviewee E4)

Regarding school food reuse policies, it was possible to identify, in an integral way, that no specific process of school food reuse is carried out. Some reports of interviewees E1, E2 and E4 illustrate such a perception.

> [ . . . ] leftover goes away, from one period to another. If there are leftovers in the morning, we don't use them the afternoon. Because the menus are different.
>
> (Interviewee E1)

> [ . . . ] nothing can be reused.
>
> (Interviewee E2)

Regarding the foods preferred by students, it was possible to identify that most students really like basic meals, such as rice and beans, that is, savory foods. In the reports of interviewees E8, E10, E11 and E6 the following was identified:

> [ . . . ] it is savory food, then we have savory pies, cakes, roasts, you know . . . They like this variety a lot.
>
> (Interviewee E8)

> [ . . . ] they like broths, bean broth with a little rice, pasta, soup, regular food, an egg sandwich. But we notice that they prefer regular food; many eat fruits, when offered, the vast majority eats fruit.
>
> (Interviewee E10)

Regarding the foods that generate the most leftovers or waste, three were the most cited: salads, fish and, finally, porridge, which students tend to reject. In the reports of interviewees E15, E16, E6, E4 and E1, it was possible to perceive these rejections, which result in waste and leftovers.

> [ . . . ] vegetables, salads.
>
> (Interviewee E15)

> [ . . . ] to have less waste, because when I see that pasta with tuna will be served, my God, a lot goes to waste, the students won't eat!
>
> (Interviewee E16)

Regarding the destination of waste and leftovers, most schools dispose of waste in the common trash. Even if it is in a separate container inside the school, its destination is normal garbage collection. All that waste goes to the landfill. However, there were reports that neighboring residents take the waste to feed animals and, eventually, some schools use the waste for composting. Examples of these reports are represented by interviewees E1, E2, E4, E5, E13 and E8.

> [ . . . ] the food that is left over: we use the bucket, right, which is for cooked food, and we separate the trash, and this bucket of food, there is a resident nearby who raises chickens, so by the end of the day late he comes to get this food for the chickens, and there are days in which he takes two full buckets.
>
> (Interviewee E1)

> [ . . . ] goes to the common trash.
>
> (Interviewee E2)

As for the most appropriate alternatives for managing food waste, most managers speak of students' awareness of food waste, but none of the interviewees seemed to master the topic nor presented projects in which this could occur. What happens are different actions, according to what each manager believes or can perform at a given time. Thus, there is the perception is that the entire school community needs to raise awareness on the topic through a top-down model, including training directors, teachers, employees and, finally, students, family members and the community where the school is located. The reports of interviewee E1, transcribed below, exemplify this perception.

> [ . . . ] I think so, thinking about doing a project to encourage them, you know, that we are going to do this and also thinking about this issue of self service. The child should self-serve, the child should have the autonomy to put on the plate whatever he/she wants to eat . . .
>
> (Interviewee E1)

Some ideas to make food leftovers more profitable were identified, from bartering to selling fertilizers and ornamental seedlings, in addition to zero-waste projects, raising awareness not only of food waste, but of all the waste that the school produces. In a way, with the minimization of waste as a whole, consequently, there is a reduction in public spending. Respondent E16 reported as follows:

> [ . . . ] they (students from the Federal University of Santa Catarina) are doing zero waste projects, and the professors are talking a lot about the student putting on the plate what he/she will eat, because leftovers will end up in the trash . . .
>
> (Interviewee E16)

Regarding the existence of awareness campaigns on food waste, there were a few reports, demonstrating how much school units need to evolve in this aspect. On the other hand, it was possible to perceive the realization of a campaign that obtained significant return, referring to the weighing of waste, which indicates that this is a good way to continue. Likewise, campaigns with the city's outsourced company responsible for garbage collection may be a positive way of maintaining a partnership with schools. In this sense, some reports by interviewees E12 were transcribed.

> [ . . . ] I think that weighing the waste was more effective, you know! When we publicized there, how much had been wasted, this was very effective. Another thing that we got really hard on a few years ago was the ingestion of candies, lollipops, these things, right, because this little piece of candy and lollipop wrap ended up flying around the school, so in addition to doing them super harm, right, it generated a lot of residue, so we started to show them what was gathered from the patio and that it doesn't matter if they put it in the dustbin, the little paper wrap flies from the trash and goes to the school space and we started showing it on the transparent glass, so when they effectively see the production of waste, the production of garbage, I think it is very effective . . .
>
> (Interviewee E12)

Allusive to the suggestions for actions to reduce food waste, the ideas were focused on educational projects to raise awareness among students, from weighing waste to establishing partnerships with nutritionists and universities, as can be seen in the statement of interviewee E10.

> [ . . . ] teachers work with children on the issue of food pyramids. Mainly in this room, which is the science laboratory, he works on the food pyramid, on the need not to waste food, the need for you to have a varied diet and enjoy what is available, at that moment.
>
> (Interviewee E10)

As for the difficulties related to the management of school meals, most of the interviewees highlighted the difficulties related to the delivery of food kits due to the pandemic,

when all schools had to adapt to different ways of managing food. There were also reports about the behavior of parents in relation to food and about the awareness of the entire school community on the issue of food waste because everything revolves around education. Some reports, in this sense, stood out, such as those of the interviewees E14, and E8.

> [ . . . ] Currently, it would be the issue of kit deliveries. In other words, there is a certain amount of food to be delivered to children, for families to be able to eat. Families need to come multiple times to make an appointment at the school to collect it. What often happens is accumulation, that is, the family does not come to collect a certain food and decides to wait for the other food to arrive to only then collect it, this creates a problem of food management. Thus, food may end up missing or must be thrown away because it has gone bad, because it has expired. One thing is to wait for dry foods, but when there are vegetables, breads, they have to be consumed as soon as possible. When we were having face-to-face classes, we could make use of those fruits and vegetables as soon as possible within the diet, but now this is no longer possible.
>
> (Interviewee E14)

> [ . . . ] the difficulty is to understand what food means inside the school. Everything inside the school, any professional, the cooks, the janitor, the teacher, the administrative technicians, everyone is educating these children, we are educators, cleaning staff, everyone . . . The problem is there are people who do not understand school meals as something educational, it is simply food. It is necessary to work with the teachers to raise awareness that food has a purpose, right, it is not just the pure and simple delivery of food. So, I think the great difficulty is making people understand that food is a process at school, which is part of the education process.
>
> (Interviewee E8)

Not knowing the technical term circular economy is not synonymous with not adopting its practices. For many people, rethinking, rejecting, reducing, reusing and reusing are institutionalized routines. However, this does not mean that they imagine that such practices corroborate the internalization of the circular economy. Remember that this is an audience with a low level of education, who do not read academic journals, who do not carry out scientific research, who do not have formal higher education, and who often do not follow market trends in their sector of activity. Therefore, there is difficulty in understanding emerging technical terms, such as the term circular economy.

Regarding the circular economy concept, none of the interviewees had exact knowledge about CE. However, when explained about circularity, it was possible to identify a familiarity in understanding, as the concept is related to sustainability. Therefore, from these interviews, it was possible to disseminate a little more about CE to some key actors in education, i.e., school managers. It was possible to perceive, through expressions and reports from the understanding of what CE is, agreement between the interviewees regarding the feasibility of carrying out projects to raise awareness of the school community, introducing the circular concept, as is also shown in [42]. However, as identified in the literature, guidelines are needed to orient these actions. In this sense, interviewees E17 and E10 expressed themselves as follows:

> [ . . . ] I'm already over it, you know . . . it's a question of having to do with ecology, it's not... not wasting natural resources.
>
> (Interviewee E17)

> [ . . . ] I may have heard it, but I don't remember now.
>
> (Interviewee E10)

With regard to circular business models, we sought to adapt them to circular school models. The analysis categories follow the ReSOLVE framework, recommended by [43], as follows: regenerate, share, optimize, loop, virtualize and exchange.

The logic that was followed was the conduction of the coding and analysis of data that are associated with the level of engagement with circular practices and the presence or absence of the categories of analysis, defined a priori, as shown in Table 8. Thus, the analysis of the data, based on predefined subcategories, classified schools according to the different levels of adoption of circularity practices in the circular school model, as follows: (i) not engaged; (ii) embryonic; (iii) initial; (iv) partial; (v) advanced; and (vi) full.

**Table 8.** Circularity from the perspective of the ReSOLVE framework.

| ReSOLVE | Purposes | Codes | Actors Involved | Level of Engagement with Circular Practices |
|---|---|---|---|---|
| Regenerate | Introduction of renewable inputs; return recovered biological resources to the biosphere (compost, organic fertilizers, slurry derived from the fermentation process of leftovers and waste). | E3, E6 | Cook Nutritionist Director Secretary of Education | Partial Stage |
| Share | Shared assets (for example, cars, rooms and appliances, etc.); reuse/use of second-hand products; to prolong the life of products through maintenance, design for durability, upgradeability, etc., emphasis on household items for sporadic use and surplus food in schools. | Not identified | Cook Nutritionist Teachers Director Secretary of Education | Not engaged |
| Optimize | To increase product performance/efficiency, to remove waste in meal preparation processes, to use smart devices to count students entering school to prepare meals only for those present. | Not identified | Cook Nutritionist Teachers Director Secretary of Education | Not engaged |
| Loop | To remanufacture products or components, to recycle materials, to use anaerobic digestion and to extract biochemicals from organic waste. | E3, E4, E6, E12 | Cook Nutritionist Teachers Director Secretary of Education | Partial |
| Virtualize | To directly dematerialize (e.g., rice leftovers from one meal become savory rice cake in the next); to indirectly dematerialize (e.g., online shopping etc.). | Not identified | Cook Nutritionist Teachers Director Secretary of Education | Not engaged |
| Exchange | To replace unused materials (for example, excess of one type of food in a school for another and to introduce balanced food that respects cultural regionalism and food preferences of each region to minimize waste); to exchange nutritious and tasty recipes between schools; to exchange positive experiences between cooks and nutritionists; to encourage healthy eating through experience-based teaching and to promote local incentives and class-based campaigns to integrate those who are most resistant to consuming a variety of foods; to substitute the current serving system with self service. | E3, E4, E8, E9 | Cook Nutritionist | Partial |

With regard to the regenerating aspect, it was identified that some schools make use of leftovers, for example, reformulating food to become something else. In schools that carry out a new proposal for food preparation when it is not well accepted by students, as in the case of tuna, nutritionists and cooks think of a new way to offer the food so that there is better acceptance. When even with the reformulation of the food there is still resistance

on the part of the students, the cook reports to the nutritionist to take actions related to a new menu offer.

> ... let's take a tuna pie for example, there is a group of students who do not have much acceptance for this item, and when it happens, then this information is passed on to the nutritionist who rethinks this preparation and considers new forms of serving the food.

(Interviewee E3)

Another aspect related to the regenerating item is that ready-to-eat foods are not used; only unprepared foods, such as fruits and vegetables, which spoil more easily.

> ... Not when it is already prepared. This is the guideline ... for example, if the tomato goes bad, if it is very ripe, it we blend them altogether and freeze it, it becomes tomato pulp. The banana, papaya, pineapple are frozen and turn into juice later.

(Interviewee E3)

Still regarding the regenerating aspect, it was very evident that the use of leftover food is not allowed, not even in the pan, even though it is in perfect condition to feed more people. In this sense, the report of participant E6 shows that there is a way to use this food more efficiently.

> ... We even they can't use it anymore, okay... but it's an amount that you can use to make a do it here on our own. If the municipality finds out about this, I can be charged somehow. But it's like this: the girls in the kitchen make a pot of sauce to make a risotto, and they mix this sauce with rice, but if the rise is over and everyone has already eaten, that sauce, theoretically savory dumplings, so we don't throw it away!

(Interviewee E6)

Regarding the cycle item, that is, schools that have an organic garden and make use of organic waste, it was observed that only four of the seventeen schools, whose directors were interviewed, have gardens and compost bins activated. The participants of these schools were E3, E4, E6 and E12. However, it is worth remembering that the interviews were carried out during the COVID-19 pandemic; nevertheless, even though some interviewees affirmed to be making use of the composter before the pandemic in 2019, at the time the survey carried out, it was no longer in use. However, they claimed they intended to start reusing it after things returned to normal. The reports indicate that it is necessary to have a team from the education department that encourages this demand and closely monitors these educational and sustainable projects in schools.

Related to item swapping, the school directors mentioned in their interviews that they exchange food with each other, and it is possible to see food redistribution from one school to another, which was confirmed by interviewees E3, E4, E8 and E9. It was advocated that this is a practice that must occur among all educational units, as it is a work carried out by nutritionists, and each nutritionist is responsible, on average, for twelve schools, and participates in constant communication via WhatsApp groups.

Sharing platforms are being adopted at the embryonic stage, with a long way to go towards the circular economy. In the same vein, the analysis by [44] showed that educational projects should incorporate circular design thinking in the education system, whether in higher or basic education. The important thing, in this sense, is to seek a systemic change that encompass the circular economy. Within this perspective, it is necessary to invest in the continuous professional development of teachers, according to the concept of circularity.

In view of the recommended scenario, the opportunity to use food in educational units is viable, and can engage in programs such as Mesa Brasil, a national food and nutrition security program to combat hunger and food waste, whose aim is to improve conditions of life in the community. The Sesc institution, which created this project, promotes, with

intertwined objectives, training courses and the integral reuse of food, as well as handicraft workshops, garbage recycling and a solidarity exchange club.

Thus, food that is left in the pan, in excellent condition for consumption and that will no longer be consumed by the students, can be donated to people who are in a situation of vulnerability through the Mesa Brasil program, which speeds up the collection of food and leads to its distribution or, if it is in large quantities, makes arrangements for food storage to better distribute it.

In view of the data collected, the considerations of [45,46] are confirmed, who affirmed that food waste is a global problem, which requires effective actions to reduce it. As such, communication is needed, not only for policymakers, but also for other stakeholders, such as schools, business and NGOs who are looking to implement new educational campaigns regarding food waste, providing consumers with a realistic perception of the problem. There is a trend towards upstream actions and policies to prevent food waste [47] and, in this context, it is evident what they indicate [46] that collaboration between the government, private sector, researchers and educational institutions is fundamental to promoting adherence to circularity aimed at waste management, in a multidisciplinary perspective [47], with joint efforts of entities [48]. Furthermore, this is in line with the statements of [26], who mention that the lack of information, misinformation and disbelief about food waste and waste recovery are among the main factors preventing supply chain constituents from connecting with each other to save edible food from landfills.

## 4. Discussion

The evidence from this study points to visible inefficiencies in the management of food waste in schools. Sending food "from the pot to the trash can" signals the inefficiency of managing food waste. Thus, the opposite of the current management model, known as food waste management, is urgent and necessary. The economic amounts "thrown away" due to the inefficiency of the current food waste management system indicate a value of BRL 1545.00 per day. This shows the absurdity that prevails. These data become very real and end up certifying the reason why, in 2018 alone, 1.3 billion tons of food were lost on the planet, about 30% of the total food produced. These data bring a huge concern, considering the need to rethink the forms of production and consumption due to the limitation of natural resources. According to [49], food waste is increasing worldwide, requiring urgent and necessary actions to mitigate this practice. In addition, challenges regarding food demand, due to the increase in the population on the planet, become increasingly urgent, putting a brake on food waste.

In developed countries, awareness and prevention are particularly important at the level of consumption, which is where food waste mainly occurs [50]. In schools, this awareness shows promise, as children understand the consequences of food waste and respond positively to the possibility of recycling food [51]. In this sense, due to the ability to transmit eating habits, public school canteens represent a unique scenario in terms of managing available resources in a sustainable way. Furthermore, alternatives for managing food waste, supported by circular supply and the ReSOLVE framework, are possible and plausible. This is what the study by [52] shows, highlighting the windows of opportunity for digital education as a contribution to the transition from the linear economy to the circular economy.

Thus, it is possible to identify that there is still enormous potential to be explored in order to integrate education and technology. In this sense, a range of options open, such as the recovery of energy from resources through different technologies. A study on circular business models in Brazilian companies, conducted by [53], showed the predominance of companies related to the service sector, which, above all, offer process virtualization, sharing, ecological products, social responsibility and emphasis on recycling.

To eradicate food waste, this study recommends the preparation of a strategic management plan for food waste. There are several actions that involve the entire school community in favor of the mitigation of waste within educational units. Furthermore, as

already emphasized, the transition to EC requires affirmative action by the government, companies and consumers, based on general awareness raising [54].

In this sense, it is possible to perceive that these are not so extraordinary actions that require high investments, such as the following:

- Think before picking up any food to consume.
- Be committed to helping take care of the school with all types of waste.
- Help monitor school cleanliness.
- Present seminars on reuse.
- Carry out collection, in their homes, of recyclable materials for use in repair practices at school.
- Help with waste sorting.
- With the support of teachers, students should carry out research on the subject.
- Help with actions, avoiding sending waste to landfills.

In any case, investments are necessary and will have a return in the long and medium term, such as, for example, investment in scales for weighing food waste. These are actions that have already brought results and that need to be applied.

Another example is investment in composters and school gardens. These are investments that will bring enriching returns to the community. In addition, there are actions involving teachers' lesson plans, namely actions that can stimulate knowledge about sustainable actions. It is also important to emphasize the importance of partnership with Mesa Brasil, so that, when it is not possible to avoid waste, ways to use food are found, for example, helping people in a state of vulnerability.

Today's researchers have the responsibility of encouraging entities to alert the population about the real situation of planetary boundaries. Awareness about what is happening should be generated because students are the future generation and they will be the ones who will suffer the most consequences. If this urgent stimulus does not occur, in the future, there will be demands from society for impartiality in the face of tragic facts that may occur and that, even today, already happen.

Another aspect that is possible to obtain with the strategic plan is the engagement of students as a natural consequence, as long as managers, teachers and employees have the same sustainable vision within schools. Sustainable practice, occurring harmoniously within the educational unit, will certainly lead to good practices in students' homes, where they will also generate cultural change and, consequently, change in society.

*Implications*

This study offers important contributions, not only to municipal schools, but also to public policies, public managers and educational institutions, especially with researchers who are supporters and sympathizers of the circular economy theme. In this sense, some opportunities arise, as follows, and allow the creation of real and applied solutions, based on the research carried out:

(a) The diagnosis that highlights the panorama of food waste can serve as a parameter to generate real and applied solutions for the management of food waste. Adherence of schools to the Mesa Brasil Program, as well as making leftovers available to homeless people around the school, are important alternatives that contribute to generating referrals to the units of analysis surveyed.

(b) The monetization of the costs of leftovers and food waste shows how much impact they have economically. These are substantive values that show that intervention is urgent and necessary to minimize these losses. Being clear about the existence of the problem is the first step in generating alternatives and solutions to circumvent this real problem, which will develop student awareness campaigns to avoid wasting food and provide ways to access food that respect children's tastes, contributing to the reduction in food waste.

(c) The inclusion of assumptions of circular business models in schools can serve as a stimulus, insight and alternative to rethink the current model of action. New alternatives, new ways of preparing food and the reuse of food (for example, rice was served that day

and from the leftovers, rice balls were served the next day) are important alternatives to reduce food loss.

(d) Circularity from the perspective of the ReSOLVE framework is also an important insight, which should serve as a test for school managers, cooks and nutritionists to rethink their current operational strategies. It is a fact that they are inefficient with regard to the management of food waste, but a municipal public policy can be created, supported by the ReSOLVE framework, to think about real solutions that have not been not adopted so far.

(e) Teaching institutions and researchers can help the municipal government in the process of measuring food waste in schools and in thinking about mechanisms to forward real solutions so that food waste is substantially reduced. New scientific studies are welcome to sharpen creativity and contribute to the school community, with alternatives that can transform current indicators of food loss and waste.

(f) Recognizing that food losses and waste are not viable ways to deal with public resources is the first step towards success in transforming the school space, which deals with food preparation in schools. Breaking resistance and circumventing bureaucratic barriers and stubborn people are other important steps towards the success of a food waste management policy.

(g) Contributions to innovation and circularity in schools: this research highlighted a more panoramic scenario about the process of school feeding in the city of Florianópolis, understanding that it is the first study carried out in schools from the perspective of EC.

In this way, it is possible, based on the collected data, to make decisions and suggest actions that contribute to the mitigation of food waste. The measuring results were positive, as in following the example: before the start of the weighing process, two buckets of waste were sent to the landfill; after the weighing process, this volume was reduced to five wasted dishes. A substantial reduction was observed, from twenty kilos of wasted food to two kilos of waste, and these two kilos were reused in the compost bin, where they were transformed into fertilizers for the garden or for marketing.

Florianopolis municipal schools can become a reference for other educational units. It is seen that the transition process to EC is a gradual process; however, the first step has been taken, and the survey of the scenario has been carried out. It should be noted that the relationship between universities, schools and the municipal education department is always important for the change in management processes.

(h) Gains/learnings generated for the academic environment with the development of this work: with regard to gains and learnings, it was possible to retain knowledge about the day-to-day of educational units and how challenging the scenario is, especially when the objects of analysis are public institutions, where practically the entire process depends on tenders, even more so in this rare, but complex, period of the COVID-19 pandemic.

Within this purpose, it was possible to perceive managers engaged in the cause to have more sustainable schools. It was also possible to perceive managers who are not concerned with the process of food waste. It is impossible to not mention the restlessness of the researcher when identifying the neglect from an educator. In any case, it is considered that everything is part of the learning process because the follow-up focused on waste management, which goes far beyond the food menu; it covers the feeding process from before preparation, during preparation and post-preparation, that is, until the student finishes their meal and goes home, in the sense that all waste generated within the school needs to be avoided or treated.

## 5. Conclusions

This study seeks to understand how food waste management is carried out in schools from the perspective of the circular economy. Evidence points to a waste of approximately 257 kg, which represents around BRL 7876.70. The mapped evidence allows the measurement of the rest of the intake (relationship between what is left on the plates and the amount of food served), which in most schools represents around one fifth of the meal served. This

indicates that the menu served to children does not always meet their expectations, which implies unconsumed food and the generation of food waste.

Little evidence of circularity assumptions has been mapped. Emphasis is given to the exchange of surplus food between schools and adherence to food-sharing platforms.

The practical contribution of this study is associated with a projection the estimates of food waste during the preparation of food, during the meal and after the meal, in addition to the economic monetization of the representative of waste for the public coffers. The theoretical contribution is associated with the generation of indicators alluding to food waste and leftovers in municipal schools, which can serve as a subsidy for forwarding public policies and more efficient alternatives for managing food not consumed by children in schools. The managerial implications of the study are associated with the possibility of creating a plan for the strategic management of food waste. This can make use of the assumptions of the circular economy to generate a perspective of re-signification of foods that are in conditions suitable for consumption.

The practical implications are the generation of a compendium of useful and relevant data for rethinking school-feeding management strategies. The current model has shortcomings, but there are simple, cheap and easy-to-operate alternatives that can have a positive and significant impact on reducing waste and leftovers. As an example, the training of school lunch and nutrition workers is mentioned to provide them with a diversified arsenal of alternatives for managing school lunches, taking into account regionalisms, cultural aspects and personal preferences of students, as well as alternative ways of using and preparing food. Another alternative is the introduction of the self-service system for school lunches. Campaigns to encourage and engage students to learn about a variety of foods, nutritional aspects and different ways of preparation are also important. Partnerships with the Mesa Brasil Program, a Federal Government program that allocates food leftovers to people in socially vulnerable situations, could lead to positive outcomes when promoted by schools. Debate between health surveillance, the state government and schools to think about solutions that value food safety, and the best use of food could provide further support for such changes, in addition to using sharing platforms, such as Olio, YoNoDesperdicio, Spoiler Alert and FoodMesh to manage leftovers.

This study presents a few limitations, such as the lack of official parameters for measuring the volume of food wasted per school. This generated the need to carry out approximate projections, which are solely and exclusively based on indicators passed on by the cooks and nutritionists, as well as on on-site observations. As an opportunity for future research, replicating the study while considering other educational contexts and countries is suggested, so that it is possible to identify a portrait of the profile of food waste in different organizational contexts. The introduction of the correction factor, the dirty leftover and the clean leftover as additional elements is recommended to deepen the analysis alluding to food waste.

**Author Contributions:** Conceptualization, S.S. and L.G.; methodology, S.S. and L.G.; software, S.S. and L.G.; validation, S.S. and L.G.; formal analysis, L.G.; investigation, L.G.; resources, S.S.; data curation, S.S. and L.G. writing—original draft preparation, S.S., L.G., F.S., C.M., S.V.S., J.B.S.O.d.A.G. and T.P. writing—review and editing, S.S., C.M. and T.P.; visualization, S.S., L.G. and T.P.; supervision, S.S.; project administration, S.S.; funding acquisition, S.S. All authors have read and agreed to the published version of the manuscript.

**Funding:** This research was funded by National Council for Scientific and Technological Development, edital 001.

**Institutional Review Board Statement:** Not applicable.

**Informed Consent Statement:** Not applicable.

**Data Availability Statement:** All data generated or analyzed during this study are included in this published article.

**Conflicts of Interest:** The authors declare no conflict of interest. The funders had no role in the design of the study; in the collection, analyses, or interpretation of data; in the writing of the manuscript; or in the decision to publish the results.

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
