# Peer review of "Management Food Waste in Municipality Schools: An Analysis from a Circular Economy Perspective"

_logistics, 2023_

Round 1

Reviewer 1 Report

In this work authors considered the perspective of food wastage associated with consumption of lunch in schools. For the same various means are used to quantify the wastage in terms of volume being produced, associated costs etc. Although outcomes of this study is in pursuit to reduce food wastage and explore possibilities of shoring it towards circular economy practices. Some of comments to author are enclosed below:

1.      In Introduction section of manuscript line number 19-20, ‘Producing food demands consumption of natural resources on an exponential scale, such production allows obtaining healthy food, recognized as nutritious and capable of satisfying the hunger of the most diverse species. This sentence needs to be fragmented to depict its intended meaning and improve ease of readability.

2.      Its being suggested to authors that avoid usage of words like ‘we’, ‘our’. For instance, it is being written as ‘We adopted the premises of the inductive approach and started from’. Further as, ‘We collected and analyzed data’. It is being repeated. Kindly, check for same thoroughly in whole manuscript.

3.      In line 567 it is mentioned that ‘none of the interviewees had exact knowledge about CE’. But then statements are there where aspects of the CE based practices is detailed. This needs to be reassured whether the detailed measures are based upon interviewers’ insights or they are tailored to suit manuscript title.

4.      Authors have simply focused on the quantification of figures of food wastage in section 4 of this manuscript. But, not detailed how this food wastage impact the procurement behaviors, its distribution and measures to curtail it. For the considered case of school students, the count is fixed. Then why this wastage is increasing also becomes a prime concern to be elaborated.

5.      It seems from the manuscript authors are more concerned for quantification of food wastage volumes but little towards its eradication and determining its cause. Hence, such perspectives also need to be elucidated in manuscript.

6.      It is being cumbersome to understand meaning of this sentence ‘adherence to food-sharing ….’ (line 709).

7.      Work implications needs to be updated in this manuscript as they are not much concrete and robust enough. Authors are requested to incorporate it as a separate heading.  

Reviewer 2 Report

The aim of the paper is the understanding of food waste management of lunches from a circular economy perspective in municipal schools and build an inductive interpretive theory using questionnaires and extended interviews. The loss estimates and their monetization are presented to propose alternatives based on ReSOLVE framework by raising the stakeholder engagement and awareness for the success and adoption of circularity practices.

 General comments:

1.)The title of the manuscript «Management food waste: an analysis with circular economy perspective« is too general since the food waste was analysed in school environment. The proposed title was already used in the referenced publication »[28] Food waste management: An analysis from the circular economy perspective«

Since there is only the difference of two words, I recommend the alignment of the title with presented research.

2.)According to the period of the data collection from October 2021 to January 2022, during the Covid-19 pandemic situation and circumstances, is not adequately described whether the schools were all the time having the average number of meals per day shown in Table 3 What was the influence of the absence of students to the number prepared meals. Does the amount of waste used in the study refer to the whole period and was estimation based on any documentation collected? When the interview took place?

 3.)Section Results is too long, the results as amount of waste and monetization are poorly explained and it is not entirely clear from the descriptions what they represent. Tables are very long, are opaque, data and descriptions that refer to all participants are without the need repeated.

Table 5:

In Third column for E1 to E4 authors write »The quantity was never measured  according to the… », from  E6 further is used »It was never measured by the …« In last, 7th column is for Participants E2 to E17 most of descriptions the same and repeated:

-        valued at R$ 9.00 per unit, there is a daily waste of meals, valued at R$ 1.80 per student.

-        amount of food waste (50g per plate), although it is in previous column amount of food waste in the meal »On average, 50 grams per student«

-        … considering that 50% of the students waste food.  

According to the same information it is recommended to shorten Table 5. Authors can provide this information only once for all participants outside the table.

Alignment of values for monthly amount in column 2 and text need to be unified.

Explain the values for monthly amount* in column 2 and monetary representation of school daily food waste in column 7. 

 Table 6 and Table 7 show monetization of leftovers and waste and total leftovers with calculations which is difficult to identify the relationship between them and get the idea about their influence on the total budget of school. I believe that monetization in such details is not very important, as there are mostly estimates of leftovers based on interviews without having measured amounts. Where there any data collected on daily basis of leftovers from responsible persons?

Table 8:

In description of Optimize and LOOP in column Actors involved authors can only list them and leave out »Should be …”

4.)In Discussion and Conclusions are summarized amount of waste and values for which is not specified to what they refer:  

-        L674-676: »The economic amounts “thrown away” due to the inefficiency of the current food waste management system indicate a value of R$ 1,545.00 per day.«   - Which food waste management system is meant? Is it in one school, or …

-        L701-702: »Evidence points to a waste of approximately 257 kg, which represents around R$ R$ 7,876.70. – From where is this waste evidenced, summarized?

 Specific comments:

 Check the manuscript for unified usage of terms, some of them are:

L585 :    change cycle to loop in the text »… regenerate, share, optimize, cycle, virtualize and exchange.«

L 627: also change to loop in »Regarding the cycle item …

ReSOLVE  - unification in  L689 ReSolve

Table 8 – MNEC -  abbreviation for ?

L695: … cuducted by [49], correct

L702: …R$ R$ - ?

Reference is not available: [24]. Prefeitura de Florianópolis. Available online: http://www.pmf.sc.gov.br/entidades/educa/index.php?cms=leis+federais+e+mu-nicipais (accessed on 22 November 2022). - PÁGINA NÃO ENCONTRADA, PAGE NOT FOUND

Author Response

Dear reviewer, please see the arrachment.

Reviewer 3 Report

The paper is really well written and structured. I have only some few suggestions:

-        -  The abstract misses the presentation of results and the main conclusions deriving from the study.

-          - Line 69 Authors should better ride the guidelines for citing references, in the sentence it is missing the subject, even if the sentence is [10], before putting the reference number, authors should write the name of the Author as a subject of the sentence. Please correct this mistake all over the paper.

-        - In the discussion section I would add more discussion of results comparing that to other situations that are different from the Brazilian one, so as to compare this situation with other in an international perspective and give a breath of internationality at the study.

Author Response

(The authors gave the same response as above.)

Round 2

Reviewer 2 Report

Dear Authors,

The review of the manuscript was not fully and properly addressed in all comments, there are still some inconsistencies in the descriptions which were not particularly exposed but are the authors responsibility to correct them in the latest version. Some of unsolved issues are exposed below, but there are still some other minor things for which is recommended to carefully read the manuscript again and correct them.

1.)Comment 3) from the review was not addressed, for which there is no explanation why the authors didn’t shorten the amount of repeated text in Tables in Section Results. As I can see there is no comment to that in the cover letter.

2.)Authors should unify all the data through the manuscript because they are using two different uses of commas and dots by monetary presentations of waste in BRL. For example: there are dots for values of thousands and comma for decimal part in Table 5, Column 2: R$ 4.865,62, and quite the opposite is use of them in Table 6, column 8: R$ 1,260.00.

3.)Why is a new, added content included twice, first time in Lines 127 – 135, and the second time in lines 587-594.

4.)Table 8 – MNEC – is acronym for what? You need to write the definition.

5.)L613: »… namely: regenerate, share, optimize, cycle, virtualize and exchange. .«  - change cycle to loop as it is used in table 8 when describing ReSOLVE framework

6)There are some other inconsistencies, like Codiname in Table 3, column 1 or just # in Table 5 Check the manuscript again for this kind of minor mistakes.

7.) It is not convenient to provide to readers a reference without any information available, as it is not  accessible on the web, but it is written in Lines 112-113 that it is not updaded (… in November 2014 and no longer updated on.) I suggest to remove it or explain why is not accessible with old data. Reference: [24]. Prefeitura de Florianópolis. http://www.pmf.sc.gov.b/entidades/educa/index.php?cms=leis+federais+e+mu-nicipais (accessed on 22 November 2022). - PÁGINA NÃO ENCONTRADA, PAGE
